# The PB2 Polymerase Host Adaptation Substitutions Prime Avian Indonesia Sub Clade 2.1 H5N1 Viruses for Infecting Humans

**DOI:** 10.3390/v11030292

**Published:** 2019-03-22

**Authors:** Pui Wang, Wenjun Song, Bobo Wing-Yee Mok, Min Zheng, Siu-Ying Lau, Siwen Liu, Pin Chen, Xiaofeng Huang, Honglian Liu, Conor J. Cremin, Honglin Chen

**Affiliations:** 1State Key Laboratory for Emerging Infectious Diseases, Department of Microbiology, and the Collaborative Innovation Center for Diagnosis and Treatment of Infectious Diseases, The University of Hong Kong, Hong Kong SAR, China; puiwang@hku.hk (P.W.); wjsong@hku.hk (W.S.); bobomok@hku.hk (B.W.-Y.M.); min.zheng@stjude.org (M.Z.); sylau926@hku.hk (S.-Y.L.); siwen531@CONNECT.HKU.HK (S.L.); u3508816@connect.hku.hk (P.C.); stevehxf@connect.hku.hk (X.H.); lhlotus@connect.hku.hk (H.L.); conor93@connect.hku.hk (C.J.C.); 2State Key Laboratory of Respiratory Disease, Institute of Integration of Traditional and Western Medicine, Guangzhou Medical University, Guangzhou 510180, China

**Keywords:** Influenza virus, H5N1, PB2, host adaptation, RNP, cross species transmission

## Abstract

Significantly higher numbers of human infections with H5N1 virus have occurred in Indonesia and Egypt, compared with other affected areas, and it is speculated that there are specific viral factors for human infection with avian H5N1 viruses in these locations. We previously showed PB2-K526R is present in 80% of Indonesian H5N1 human isolates, which lack the more common PB2-E627K substitution. Testing the hypothesis that this mutation may prime avian H5N1 virus for human infection, we showed that: (1) K526R is rarely found in avian influenza viruses but was identified in H5N1 viruses 2–3 years after the virus emerged in Indonesia, coincident with the emergence of H5N1 human infections in Indonesia; (2) K526R is required for efficient replication of Indonesia H5N1 virus in mammalian cells in vitro and in vivo and reverse substitution to 526K in human isolates abolishes this ability; (3) Indonesian H5N1 virus, which contains K526R-PB2, is stable and does not further acquire E627K following replication in infected mice; and (4) virus containing K526R-PB2 shows no fitness deficit in avian species. These findings illustrate an important mechanism in which a host adaptive mutation that predisposes avian H5N1 virus towards infecting humans has arisen with the virus becoming prevalent in avian species prior to human infections occurring. A similar mechanism is observed in the Qinghai-lineage H5N1 viruses that have caused many human cases in Egypt; here, E627K predisposes towards human infections. Surveillance should focus on the detection of adaptation markers in avian strains that prime for human infection.

## 1. Introduction 

Human infection with avian H5N1 subtype virus was first documented in Hong Kong in 1997, where it caused 18 infections and six deaths [1,2]. Available epidemiological data show that avian H5N1 virus has since become endemic in poultry in China [3]. Regional dissemination occurred subsequently and human infections with H5N1 have been identified in Southeast Asian countries, including Thailand, Vietnam, Laos and Indonesia, since 2003 [4]. Following a major outbreak in migratory wild birds at Lake Qinghai, China, in 2005, H5N1 virus spread further to Europe, Africa and the Middle East, leading to occasional human infections in affected countries [5]. According to a World Health Organization report, 860 human cases, including 454 deaths, have been confirmed in 16 countries between 2003 and July 2015 (http://www.who.int/influenza/). Genetic analysis has demonstrated that avian H5N1 viruses are highly diverse and prevalent with distinct regional genetic lineages becoming apparent in the past decade [4]. While human infections have been identified in several countries, two genetic variants of avian H5N1 virus have caused nearly 70% of all human infections; one, a descendant of the Qinghai-lineage virus (or subclade 2.2), is currently circulating in Egypt, Africa and the Middle East, while the other genetic variant is prevalent in Indonesia (or subclade 2.1). Long-term endemicity of H5N1 virus has led to the emergence of genetic reassortant H5N6 and H5N8 viruses, the former of which has caused human infections, and spread to more geographical regions since 2014 [6,7,8]. The H5N1 subtype avian influenza virus is the most prevalent and widely disseminated endemic panzootic virus in history. It is important to understand how the H5N1 virus gained the ability to perpetually circulate in avian species while maintaining the ability to cross host range barriers and infect humans. Although sustained human to human transmission has not yet been observed, the pandemic potential of this virus remains a concern.

In addition to adopting host-specific receptor binding through changes to viral hemagglutinin, host adaptation mutations in the basic polymerase subunit PB2 are recognized to be a critical step for avian influenza viruses to gain the ability to infect and replicate in mammalian cells [9,10,11]. Various adaptation markers in the viral PB2 polymerase have been identified in human influenza A viruses and in isolates from human infections with avian H5N1, H7N7 or H7N9 subtype viruses, suggesting that different adaptive mechanisms may be utilized in the process of cross species transmission [12,13,14,15,16,17,18]. Among the known PB2 adaptive substitutions, E627K, and to a lesser extent D701N, are the most common and well characterized in H5N1 human isolates. It has been suggested that the E627K substitution allows virus to evade host restriction of replicative activity of avian PB2 polymerases, but a detailed mechanism remains unclear [19]. A recent study showed that the host pattern recognition receptor, RIG-I, distinctively interacts with avian and human PB2 polymerases and inhibits avian virus replication [20]. Another possible mechanism associated with PB2 host adaptation involves interaction with host importin α isoforms, which mediate nuclear import of RNP polymerases [21,22]. Recent studies also suggest that PB2 E627K and K526R substitutions may be involved in regulation of virus replication through interaction with viral NEP during virus infection of cells [13,23]. H5N1 viruses characterized from poultry or wild birds generally contain avian type 526K, 627E or 701D PB2 polymerase, except for the 627K PB2 genotype, which is present in the Qinghai-like virus (subclade 2.2) that is currently endemic and causing human infections in Egypt [5]. Notably, E627K is rarely found in human H5N1 cases from Indonesia, where a K526R substitution is observed in 80% of human isolates [13,24]. K526R is a recently identified adaptation marker that supports influenza virus replication in mammalian hosts; first observed in seasonal H3N2 viruses in the early 1970s, it is now present in all currently circulating H3N2 viruses [13]. It is suggested that the selection of different PB2 adaptation mechanisms may depend on the genetic lineage of avian H5N1 viruses [25]. 

Since PB2-E627K is rarely found among H5N1 human isolates from Indonesia, it seems possible that, for the genetic constellation of Indonesian H5N1 viruses, optimal function of the polymerase complex for virus replication is independent of E627K, and instead requires K526R adaptive substitution. Similarly, a specific PB2 adaptation mechanism has been observed in 2009 pandemic H1N1, which utilizes 590S/591R, rather than 627K, for replication in humans [18]. Because significantly higher numbers of human infections with H5N1 virus have occurred in Indonesia and Egypt, compared to other affected areas, it may be speculated that there are specific viral factors for human infection within avian H5N1 viruses in both locations. Egyptian H5N1 viruses exclusively contain 627K PB2, and the majority of isolates also lack the 154–156 glycosylation site in the HA protein, which enhances binding to human-like receptors; these properties may prime the virus for infection and replication in humans [26]. Because K526R PB2, but not E627K, is predominantly found in H5N1 human isolates from Indonesia, this study examined if K526R PB2 is a predisposing factor for virus infection of humans. Analysis of sequences revealed that H5N1 virus containing K526R PB2 emerged 2–3 years after the introduction of H5N1 virus into Indonesia, coincident with the emergence of human infections. We found that virus carrying K526R PB2 replicates better than virus without this substitution in mammalian cells and mice, but showed no apparent deficit in growth in avian cells. K526R substitution in PB2 appears to serve as a pre-existing adaptation in avian H5N1 virus from Indonesia and likely plays a role in enabling this variant to cross the species barrier and infect humans.

## 2. Materials and Methods

### 2.1. Viruses and Rescue of Recombinant Viruses

All influenza viruses used in this study were rescued using the reverse genetics (RG) technique to ensure the genetic homogeneity of the virus. Viruses were propagated in embryonated chicken eggs (9–11 days old) for 24 h (P1) and titered by plaque assay in MDCK Cells (American Type Culture Collection, Manassas, VA, USA). The two strains of Indonesian H5N1 viruses used in this study are a human isolate, A/Indonesia/5/05 (IND5), and an avian isolate, A/Chicken/Indonesia/2A/03 (CK2A) [4,24]. The sequences of the viruses were deposited in Genbank (Accession numbers for IND5 are ABI35999, ABI36000, ABI36002, ABI36003, ABW06108, ABW06107, ABI36004, and ABI36006; for CK2A are AY651710, AY651655, AY651599, AY651323, AY651490, AY651435, AY651377, and AY651543). The eight segments from each of the two Indonesia H5N1 viruses were cloned into a pHW2000 plasmid-system [27,28]. Additional PB2 segments mutated at position 526 and elsewhere were generated using the QuikChange® mutagenesis kit (Agilent, Santa Clara, CA, USA) and used for the rescue of PB2 mutant viruses. Full length PB2 sequences of human and avian H5N1 viruses from the NCBI Influenza Virus Resource Database, together with all influenza A H5N1 human isolates in GISAID (http://gisaid.org), were analyzed for prevalence of the K526R substitution.

### 2.2. Minigenome Reporter Assays

Luciferase activity-based minigenome reporter assays were performed as described previously [13]. The coding region of the ribonucleoprotein (RNP) complex (PB1, PB2, PA and NP) was cloned separately into pCX plasmids. HEK293T cells (ATCC, Manassas, USA) were transfected with the RNP expression plasmids, a firefly RNP luciferase reporter plasmid and a Renilla luciferase expressing plasmid pRL-TK (as an internal control), and the transfected cells incubated at 37 °C or 33 °C. Luciferase activity was measured at 24 h post transfection using a Dual Luciferase Reporter Assay System (Promega). The firefly RNP polymerase activity was normalized against the Renilla activity. For luciferase reporter assays in DF-1(ATCC, Manassas, USA), a firefly luciferase reporter containing a chicken pol I-driven promoter was constructed based on a previous report [29]. DF-1 cells were incubated at 39 °C. To study the effect of NEP on RNP activity, NEP expressing plasmid was co-transfected with the RNP expressing plasmids and the reporter plasmids, and RNP activity measured as described.

### 2.3. Virus Growth Kinetics

MDCK, A549 (ATCC, Manassas, VA, USA) and DF-1 cells were used in the virus growth kinetics study. In general, confluent cells were infected with RG virus at the indicated multiplicity of infection (MOI). The viral inoculums were removed after 1 h of adsorption at 37 °C and replaced with infection medium containing 1 µg/mL of TPCK-trypsin. Infected cells were further incubated at the indicated temperatures, supplied with 5% CO_2_. Culture supernatants were collected at different time points and viral titers determined by plaque assay in MDCK cells. MDCK and A549 cells were cultured at 37 or 33 °C, while DF-1 cells were cultured at 39 °C. 

### 2.4. Mouse Experiments and Lethal Dose (MLD_50_) Determination

BALB/c mice, aged 4–6 weeks, were obtained from the Laboratory Animal Unit, the University of Hong Kong. For MLD_50_ determination, groups of four or six mice were anesthetized by intraperitoneal injection of a mixture of ketamine and xylazine, and then intranasally inoculated with 25 µL of RG virus diluted in PBS, the dose range being 10^0^–10^6^ PFU. Body weight and survival were monitored daily for 14 days after infection. The MLD_50_ was calculated by the method of Reed and Muench [30]. To assess virus replication in mouse lungs, three mice from each group were euthanized at 72 h post infection and lung tissues from each mouse collected and homogenized for virus titration. The protocols of animal experiment and use of embryonated chicken eggs (9–11 days old) were approved by the Committee on the Use of Live Animals in Teaching & Research (CULATR), Li Ka Shing Faculty of Medicine, the University of Hong Kong (CULATR-3064-13, from 26-6-2013 to 25-6-2016). The CULATR of the University of Hong Kong follows animal welfare regulations of Hong Kong Special Administration Region legislations and AAALAC recommended standards/guidelines (http://www. aaalac.org/about/guidelines.cfm). The “Guide for the Care and Use of Laboratory Animals, 2011, NRC, USA” (http://www.aaalac.org/resources/theguide.cfm) is the main document used by the HKU CULATR. The risk assessment of this study was approved by the safety committee on BSL-3 facility and infectious agents, Li Ka Shing Faculty of Medicine, the University of Hong Kong.

### 2.5. Quantitative RT-PCR (qRT-PCR) Assays for vRNA and Viral mRNA 

To determine the expression levels of vRNA and viral mRNA in infected cells, A549 cells were infected with different RG viruses at an MOI of 1. Total RNA was extracted at the indicated time points and 1 µg of total RNA reverse transcribed using a High Capacity cDNA Reverse Transcription Kit (Invitrogen, Carlsbad, CA, USA). For detection of vRNA and mRNA, uni-12 and oligo dT primers, respectively, were used in the reverse transcription (RT) reaction. Expression of these two NP RNA species was quantified using SYBR Premix Ex Taq reagent (Takara) in an ABI 7500 Fast Real-Time PCR machine. For the detection of NP, NP-545s (5′-cagtgaaggggatagggaca-3′) and NP-746 as (5′-ccaggatttctgctctctcg-3′) primers were used. β-actin expression in each sample was used for normalization. Expression of NP genes was estimated by relative quantitative RT-PCR and normalized with the β-actin gene, in accordance with a previously described method [31]. The amplification program was as follows: 95 °C for 30 s, followed by 40 cycles of 95 °C for 5 s, 60 °C for 30 s. The specificity of the assay was confirmed by melting-curve analysis at the end of the amplification program (65–95 °C, 0.1 °C/s). Further confirmation of the specificity of qRT-PCR for specific RNA species was performed as we have described previously [13].

### 2.6. Growth Competition Assay with Different PB2 Mutant Viruses 

Mutation at the PB2 526 position was performed using the QuikChange Site-Directed Mutagenesis Kit (Agilent). Growth competition of the H5N1 avian wild type virus A/Chicken/Indonesia/2A/03 (PB2 526K and 627E) against its mutants (PB2 526R or PB2 627K) was carried out according to established procedures [13,32]. DF-1 cells were infected at an MOI of 0.001 with wild type virus and one mutant virus, mixed at a ratio of 1:1. Several serial passages were performed. Viral RNA was isolated from cell culture supernatants at 48 h post infection, and the PB2 gene amplified by RT-PCR and sequenced.

### 2.7. Immunoprecipitation (IP) Assay

To study the interaction between NP and PB2 in the RNP complex, pCX plasmids expressing PB1, PA, NP and flag-tagged PB2 were transfected into HEK293T cells together with an RNP reporter plasmid. After 48 h, cells were lysed in cell lysis buffer (Tris-HCl 50 mM, 150 mM NaCl, 1% TritonX, pH 7.4). Cell debris was cleared by centrifugation. One microliter of laboratory made polyclonal NP antibody (1:3000) was added to the cell lysate and the mixture incubated for 3 h at 4 °C. Thirty microliters of washed Protein A sepharose beads (GE Health) was added to the lysate, followed by incubation for 1 h at 4 °C. The beads were washed with the lysis buffer and finally boiled in SDS buffer at 95 °C. Proteins were resolved in SDS-PAGE and detected by Western blot using anti-NP and anti-Flag (Sigma, St. Louis, MO, USA) antibody.

### 2.8. Phylogenetic Analysis

Phylogenetic trees were constructed using the neighbor-joining method with the Tajima-Nei model of nucleotide substitution using MEGA X. All available sequences of H5N1 PB2, HA and PB1 from Indonesia were acquired from GenBank and GISAID for analysis. The bootstrap values from 1000 replicates were calculated to evaluate the reliability of the phylogenetic tree. 

### 2.9. Statistical Analysis

Statistical analysis was carried out using GraphPad Prism V5 software (GraphPad Software, Inc, San Diego, CA, USA).

## 3. Results 

### 3.1. K526R Substitution Emerged in Avian H5N1 Virus, Coincident with Human Infections 

Apart from human cases infected with Qinghai-like viruses, which harbor 627K PB2 in avian species, and Indonesian viruses the later which rarely contain 627K PB2, about 30% of human isolates from rest of the affected areas mainly from China, Vietnam and Thailand contain 627K, with the E627K substitution presumably being acquired following infection. As reported previously, about 80% of H5N1 human isolates from Indonesia contain K526R PB2 [13]. To understand if the viruses acquired the K526R substitution prior to or after human infection, we analyzed public available sequences from avian and human isolates from Indonesia. While H5N1 virus was first detected in Indonesia in 2003, it is suggested that the first introduction may have occurred as early as 2002 [33]. It appears that K526R PB2 was not present in Indonesian avian H5N1 viruses until 2005, the same year that the first human case of H5N1 infection was identified in Indonesia [4,24] (Figure 1). Phylogenetic analysis of PB2 genes of avian and human isolates showed that viruses carrying 526K and 526R separate into two clusters. It is notable the later-emerging avian H5N1 viruses that carry 526R PB2 (in blue) clustered with the majority of human isolates (in boldface) (Figure 1). It appears that the original avian H5N1 virus introduced into Indonesia has changed over time to gain a certain level of ability to infect humans. These results raise the possibility that certain adaptations in Indonesian H5N1 viruses may prime them for human infection.

### 3.2. K526R Supports H5N1 Virus Replication in Mammalian Cells

To understand if K526R substitution contributes to virus replication, we created two pairs of reverse genetic viruses based on A/Chicken/Indonesia/2A/2003 (CK2A, avian) and A/Indonesia/5/2005 (IND5, human), which originally contained 526K and 526R PB2, respectively (Figure 1). Minigenome reporter assays showed that IND5 RNP polymerase activity is significantly reduced if 526R PB2 is back mutated to 526K (Figure 2A). In contrast, RNP polymerase activity is significantly increased when the K526R substitution is introduced into the avian PB2 of the CK2A isolate. Previous studies indicate that adaptive mutations may facilitate avian virus replication at lower temperatures, favoring replication in human upper respiratory tract tissues. We found that the enhancement of RNP polymerase activity associated with 526R is consistently more apparent at 33 °C (Figure 2B). However, 526R-PB2 does not exert a similar positive effect on polymerase activity when compared to 526K-PB2 in avian origin DF-1 cells (Figure 2C). To further test the impact of K526R in virus replication, the growth kinetics of the two pairs of reverse genetic versions of viruses containing either 526K or 526R in the CK2A or IND5 backbone were analyzed. Consistent with the RNP polymerase activity result, back substitution to 526K in the PB2 of IND5 virus decreased virus replication in MDCK and A549 cells, but not in DF-1 cultures (Figure 3A). Conversely, introduction of 526R into PB2 of the avian CK2A virus enhances virus replication in mammalian cells, but this positive effect was not observed in DF-1 cells (Figure 3B). Adaptation of E627K in PB2 was found to facilitate virus replication at a lower temperature in the upper respiratory tract [34]. We also found the positive effect of 526R on virus replication in mammalian cells to be more apparent at 33 °C in MDCK and A549 cells (Figure 3C,D). Further examination of viral mRNA, vRNA and viral protein synthesis showed that 526R enhances both viral mRNA transcription and vRNA replication in single cycle of infection in A549 cells (Figure 4).

Interaction between PB2 and nuclear exporting protein (NEP) may regulate the virus–host adaption process during virus replication, although a detailed mechanism remains elusive. Multiple functions of NEP during the influenza virus life cycle have start to emerge in recent years [35]. Previous studies have shown that 627K and 526R substitutions in PB2 may promote optimal replication of virus in the presence of NEP in the backbone of H5N1 and H7N9 viruses [13,23]. We tested if K526R substitution in H5N1 viruses from Indonesia may enhance interaction between NEP and viral polymerase complex. Our results show that for RNP which contain 526R PB2, polymerase activity remains relatively stable in the presence of increasing levels of NEP, while RNP with 526K PB2 are significantly affected by increased amounts of NEP (Figure 5A,B). Interestingly, the differential effect of NEP on 526R-PB2 and 526K-PB2 RNP polymerase activity is not observed in DF-1 avian cells (Figure 5C), suggesting that interaction between PB2 and NEP may be host dependent. An adaptive E627K substitution in PB2 was found to enhance PB2 interaction with NP in the RNP complex [36]. Similarly, an immunoprecipitation assay reveals that 526R-PB2 (IND5-WT or CK2A-K526R) has a higher affinity interaction with NP than 526K-PB2 does (Figure 5D).

Taken together, these results clearly show that K526R substitution in PB2 significantly enhances virus replication in MDCK and A549 cells, but does not attenuate virus replication in avian cells, supporting the contention that, similar to E627K, K526R is a host adaption marker important in the context of viral replicative function in humans.

### 3.3. K526R PB2 Enhances H5N1 Virus Replication in Mice

We previously showed that K526R enhances avian H7N9 virus replication in mice when combined with 627K [13]. Since the majority of human isolates from Indonesia do not contain the 627K mutation, we evaluated whether these viruses have adapted to use a different mechanism, namely K526R, to facilitate replication in a mammalian host. Pairs of H5N1 viruses containing either 526K or 526R PB2, as described above, were used for infection of BALB/c mice. Infection with a series of inoculums, ranging from 10^2^ to 10^6^ PFU, showed that both 526K and 526R PB2 viruses are virulent in mice at the higher dose inoculums of 10^4^ to 10^6^ PFU, with all infected mice dying between four and nine days of infection [13]. At lower inoculums, 10^3^ PFU of IND5 virus still killed all infected mice, regardless of whether it was carrying WT-526R or back mutated 526K (Figure 6A). In contrast, mice infected with 10^3^ PFU of the avian isolate CK2A showed no disease symptoms, while CK2A-526R caused substantial body weight loss at this dose, with an almost 20% reduction in infected mice around Days 7–10 post infection, from which all mice subsequently recovered. It is notable that at the lower dose, 10^2^ PFU, WT IND5 still killed all infected mice, but mice infected with IND5 virus containing back mutated 526K PB2 showed no disease symptoms, being indistinguishable from the PBS control group, supporting the idea that the K526R substitution is critical for replication of Indonesian H5N1 viruses in mice. Mice infected with 10^2^ PFU of CK2A-526R virus showed only mild body weight loss from Day 9 post infection, and all recovered (Figure 6B). Measurement of the MLD_50_ of these virus strains with 1 to 10^6^ PFU infectious dose showed that back mutation to R526K in the human isolate (IND5) reduces virus virulence in mice (Table 1). WT IND5 exhibits higher virulence than mutated avian CK2A-526R virus, which contains K526R PB2, suggesting that other mutations (see section below), yet to be defined, may also contribute to virus replication in mice. Consistent with the body weight curve, analysis of virus titers in lung tissues obtained 72 h post infection shows that K526R substitution in PB2 enhances virus replication in mice (Figure 6C). In summary, these animal experiments further demonstrate that K526R substitution in PB2 is important for in vitro virus replication in mammalian host.

### 3.4. PB2-K526R Is Stable in both Avian and Mammalian Hosts

Acquisition of the E627K substitution in PB2 following infection in humans or other mammalian hosts has been observed for avian H5N1, H7N7 and H7N9 viruses [12]. Our results showed that the IND5 human isolate is highly virulent in mice (Table 1), even though it does not contain the 627K PB2 marker. To examine if 627K may be acquired after virus infection in mice, or if K526R is sufficient to confer virus replication ability in mice, we examined viruses isolated from fatally-infected mice died from Day 5 post infection in our experiments. Notably, no E627K or other known substitution was observed in any of the seven mice fatally infected with IND5 virus (Table 2). However, E627K substitution was found in isolates from two of four mice fatally infected with reverse-mutated IND5-526K virus, suggesting that if 526R is absent then virus may adapt by gaining E627K so that it can replicate in mice. Similarly, E627K substitution was found in one of five mice fatally infected with CK2A-526R virus, while viruses from three out of four mice infected with CK2A-WT avian isolate were found to contain this substitution. These results suggest that E627K substitution is not required when the K526R PB2 adaptation is present. To further analyze if K526R PB2 is a specific pre-existing mammalian adaptation marker in Indonesian H5N1 viruses, we compared the differential effects of 526R and 627K on replication of the CK2A avian virus strain, which represents virus in the early epidemic in poultry before emergence of human infection occurred in Indonesia. Growth kinetics of CK2A-WT or CK2A virus containing either 526R (CK2A-526R) or 627K (CK2A-627K) PB2 were analyzed in DF-1 cell infections. While CK2A-WT and CK2A-526R viruses replicate comparably in DF-1 cells, replication of CK2A-627K-PB2 virus is significantly decreased at the 14 and 24 h time points (Figure 7A). Furthermore, analysis of viruses from co-infection of DF-1 with equal amounts of CK2A-WT and CK2A-526R or CK2A-627K PB2 virus showed that populations of CK2A-526R and CK2A-526K (WT) maintain the initial ratio, but that the CK2A-627K genotype disappears after four passages (Figure 7B,C), which suggests that 526R PB2 has no detrimental effect but that 627K may somehow compromise virus replication in avian cells. It may be postulated that, for this genetic lineage of avian H5N1 viruses, 627K PB2 only functions to allow replication in mammalian, but not avian, hosts. Indeed, comparison of virus growth kinetics and population change in co-infection assays using CK2A-627K and CK2A-526R viruses showed that substitution of 627K-PB2 elicited increased replication efficiency for CK2A virus in A549 cells (Figure 7D,E). Taken together, these results indicate that host adaptive mutations in the PB2 gene enhance H5N1 virus replication in mice and that K526R PB2 is a pre-existing adaptive marker present in Indonesian H5N1 viruses currently circulating in avian species, which is necessary and sufficient to support virus replication in mammalian cells.

### 3.5. Linkage of R288Q and K526R in Indonesian H5N1 Viruses

Apart from seasonal H3N2 viruses arising since the 1970s and the Indonesian H5N1 viruses, few other circulating influenza viruses carry the PB2 polymerase K526R substitution. It was suggested that, subsequent to the 1968 reassortment event, which incorporated an avian PB1, acquisition of K526R in the PB2 gene of H3N2 enhanced the function of PB2-627K in the polymerase complex to optimize replication, and perhaps also transmission, of H3N2 in humans [13]. Uniquely, the K526R PB2 substitution in Indonesian H5N1 viruses is rarely associated with either E627K or D701N [13]. To identify other genetic linkages that might be involved in the K526R PB2 adaptive strategy in H5N1 viruses from Indonesia, we examined available sequences in public databases and found that R288Q is exclusively associated with K526R in the PB2 of all characterized Indonesian H5N1 viruses. We compared the PB2 sequences of the IND5 and CK2A isolates and found them to differ by eight amino acids (Table 3). Analysis using a minigenome reporter assay indicated that R288Q is relevant to RNP polymerase activity. To test if R288Q coordinates functionally with K526R, we analyzed RNP polymerase activity with PB2 containing the R288Q substitution in the background of the CK2A avian and IND5 human strains. While introduction of R288Q into CK2A PB2 moderately increases RNP polymerase activity, the combination of R288Q and K526R substitutions significantly enhances the positive effects on polymerase activity exerted by R288Q or K526R alone (Figure 8A). As expected, Q288R reverse substitution in PB2 from IND5 decreases polymerase activity and dual reversal to Q288R and R526K severely diminishes polymerase activity (Figure 8B). Examination of growth properties in A549 cells further demonstrated that R288Q and K526R together enhance virus replication in the CK2A background (Figure 8C). These results suggest that R288Q/K526R PB2 was selected for optimal function of RNP in Indonesian H5N1 viruses and that this constellation is independent of E627K function.

## 4. Discussion

Cross species transmission of avian influenza A virus is restricted by host barriers which may include receptor specificity, replication fitness and host innate immunity. Emerging human infections with avian influenza viruses have increased in incidence over the past 20 years [37]. Major concerns have been raised regarding the pandemic potential of the H5N1 and H7N9 subtypes in particular, which have continuously crossed host barriers to cause sporadic human infections since 1997 and 2013, respectively [1,38,39]. New genetic variants of H5N8, H5N6 and H5N2 viruses derived from H5N1 and H9N2 subtype viruses are endemic in poultry in China; some of these have spread to Europe and, in 2014, were found for the first time in North America [6,7]. Currently, there is no effective way to eradicate these panzootic viruses from avian species. Close surveillance for adaptation markers associated with human infections among these viruses is necessary for preparedness against potential emerging pandemic strains.

Historical pandemics were caused either by reassortant viruses incorporating avian viral segments or whole animal viruses [40,41,42]. The 2009 H1N1 virus is believed to have originated in swine, which may have allowed the reassortant virus to gain further adaptations in this intermediate host and increase its ability to replicate in humans prior to the onset of the pandemic in the spring of 2009 [41]. Both the 1957-H2N2 and 1968-H3N2 pandemics were initiated by the emergence of reassortant viruses, which arose from viruses that were circulating in humans and had recently acquired avian viral genome segments [43,44]. The mechanism for the genesis of the pandemic 1918 virus is less clear, with one hypothesis suggesting that it may have arisen from an avian-like influenza virus, while a genetic dating analysis study suggests the possibility that reassortment events formed this virus [42,45]. If direct cross species transmission of virus from avian species to humans can occur, it seems reasonable to suggest that certain host-specific adaptations must pre-exist in some avian influenza viruses, and that these adaptations do not impair virus replication fitness in avian species, ensuring virus prevalence. There are concerns that the unprecedented long term endemicity of H5N1 virus in poultry and continuous spillovers to humans may provide an opportunity for the virus to accumulate sufficient adaptations which facilitate human to human transmission. Host adaptation markers for receptor binding and replication abilities are key components which may determine the potential for cross species transmission of an avian influenza A virus. Indeed, two laboratory adaptation experiments using different H5N1 virus strains demonstrated that only a few further mutations in the hemagglutinin may be needed for a virus to gain the ability to transmit between ferrets [9,10]. Apart from gaining host receptor binding specificity, adaptation of PB2 polymerase is also vital for viruses to become able to replicate in humans [12,46]. Influenza A viruses have employed different adaptive strategies involving mutations in the PB2 subunit of the polymerase complex to gain replication ability in mammalian hosts [13,47]. E627K is the most well recognized adaptation marker for human infections caused by avian H5N1, H7N7 and H7N9 viruses. The clade 2.2 Eurasian-lineage H5N1 virus, also referred to as Qinghai-like virus, contains a mammalian type 627K PB2, while maintaining replication fitness for circulating in avian species [25]. It is possible that this “pre-adapted” 627K PB2 conferred this clade 2.2 avian H5N1 virus with the ability to cross the species barrier to infect humans. Similarly, a D701N adaptation in a duck H5N1 virus which predisposed towards transmission to a mammalian host has also been reported [15].

Indonesia has the second largest total number of H5N1 human cases, with 199 recorded since 2005. However, most human H5N1 isolates from Indonesia do not harbor the E627K or D701N substitutions frequently observed in other human isolates from avian H5N1 or H7N9 virus infections. We previously reported that the K526R substitution enhances the effect of E627K in promoting replication of H3N2 and H7N9 viruses, and that K526R is present in 80% of H5N1 human isolates from Indonesia [13]. Here, we propose that K526R is a pre-existing mammalian adaptive marker of avian H5N1 viruses in the Indonesia lineage and provide the following supporting evidence: (1) K526R is rarely found in avian influenza viruses but was identified in H5N1 viruses 2–3 years after the virus emerged in Indonesia, coincident with the emergence of H5N1 human infections in Indonesia; (2) K526R is required for efficient replication of Indonesia H5N1 virus in mammalian cells in vitro and in vivo and reverse substitution to 526K in human isolates abolishes this ability; (3) Indonesian H5N1 virus, which contains K526R-PB2, is stable and does not further acquire E627K following replication in infected mice; and (4) virus containing K526R PB2 shows no fitness deficit in avian species. The contention that the gene constellation of K526R harboring virus is stable in the mammalian host, without needing to acquire other known adaptations, was also observed in a previous study where A/Indonesia/5/2005, which contains only K526R but no other known adaptive markers, did not acquire E627K or other known adaptations in PB2 even after being passaged 10 times in ferrets [9].

The functions of K526R-PB2 in H5N1 virus from Indonesia and E627K in H5N1 virus from Egypt to predispose towards the ability to replicate in humans suggests that mammalian host adaptive mutations can be stably acquired by avian influenza viruses and maintained in poultry. These adaptations do not hinder replication fitness of virus in avian species but provide H5N1 viruses with panzootic potential. Laboratory adaptation of H5N1 viruses in ferrets showed that only a handful of additional substitutions in the HA is sufficient to allow H5N1 viruses to transmit between animals, indicating that at least some genetic variants of currently circulating H5N1 viruses are likely able to replicate in humans but are merely restricted by receptor binding specificity [9,10]. Because the H5N1 virus is still primarily confined to replication in avian host, maintaining avian type receptor preference is necessary for virus circulation, hence the chance of a virus with human-like receptor binding preference becoming prevalent in an avian host is very unlikely. However, avian influenza viruses containing pre-existing genetic markers of mammalian adaptation for viral replication function, such as K526R and E627K in the PB2 polymerase, are equally as likely to become prevalent, as evidenced by the subclade 2.2 (Egyptian) and 2.1 (Indonesian) H5N1 viruses. It is likely that there are multiple host adaptive strategies for adaptation to mammals for PB2 of the avian influenza viral polymerase and also involve other viral functions including NS1, NEP and NP [13,21,23,47,48,49,50]. We propose that the PB2 K526R substitution primes avian H5N1 viruses from Indonesia for infecting humans. For these avian viruses with a pre-existing adaptation, any further adaptation within a mammalian host may transform the virus into a fully human virus. The pandemic potential posed by the currently circulating avian H5N1 influenza viruses cannot be ignored. Preventive measures should be considered to reduce or control endemicity of H5N1 viruses carrying these markers in poultry, to minimize the chance of transmission to humans.

## Figures and Tables

**Figure 1 viruses-11-00292-f001:**
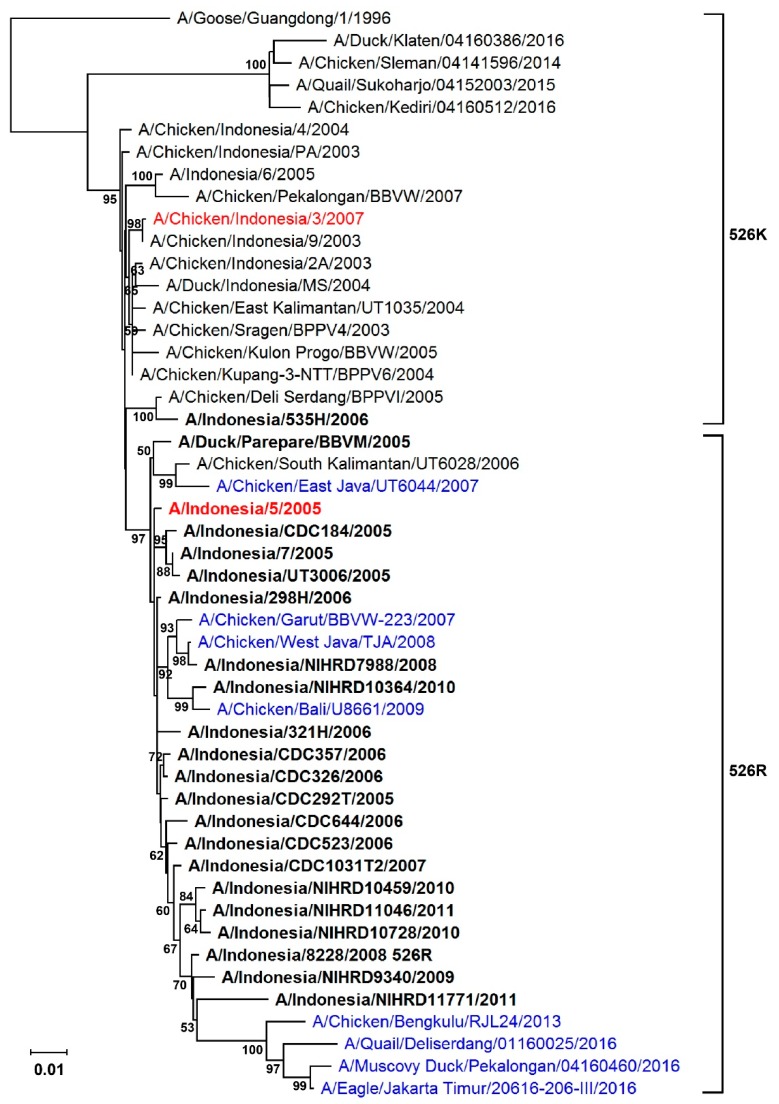
Phylogenetic tree of the PB2 gene of Indonesian H5N1 virus. The tree was constructed using the neighbor-joining method with the Tajima-Nei model of nucleotide substitution using MEGA X. Blue indicates avian strains carrying Arg at position 526 of the PB2 gene (526R). Human strains are highlighted in boldface. Numbers represent the bootstrap values (percentages) from 1000 replicates: only bootstrap values greater than 50 are shown. Strains in red are the avian isolate, A/Chicken/Indonesia/2A/2003 (CK2A), and the human isolate, A/Indonesia/5/2005 (IND5), used as backbones for constructing reverse genetic versions of viruses in this study.

**Figure 2 viruses-11-00292-f002:**
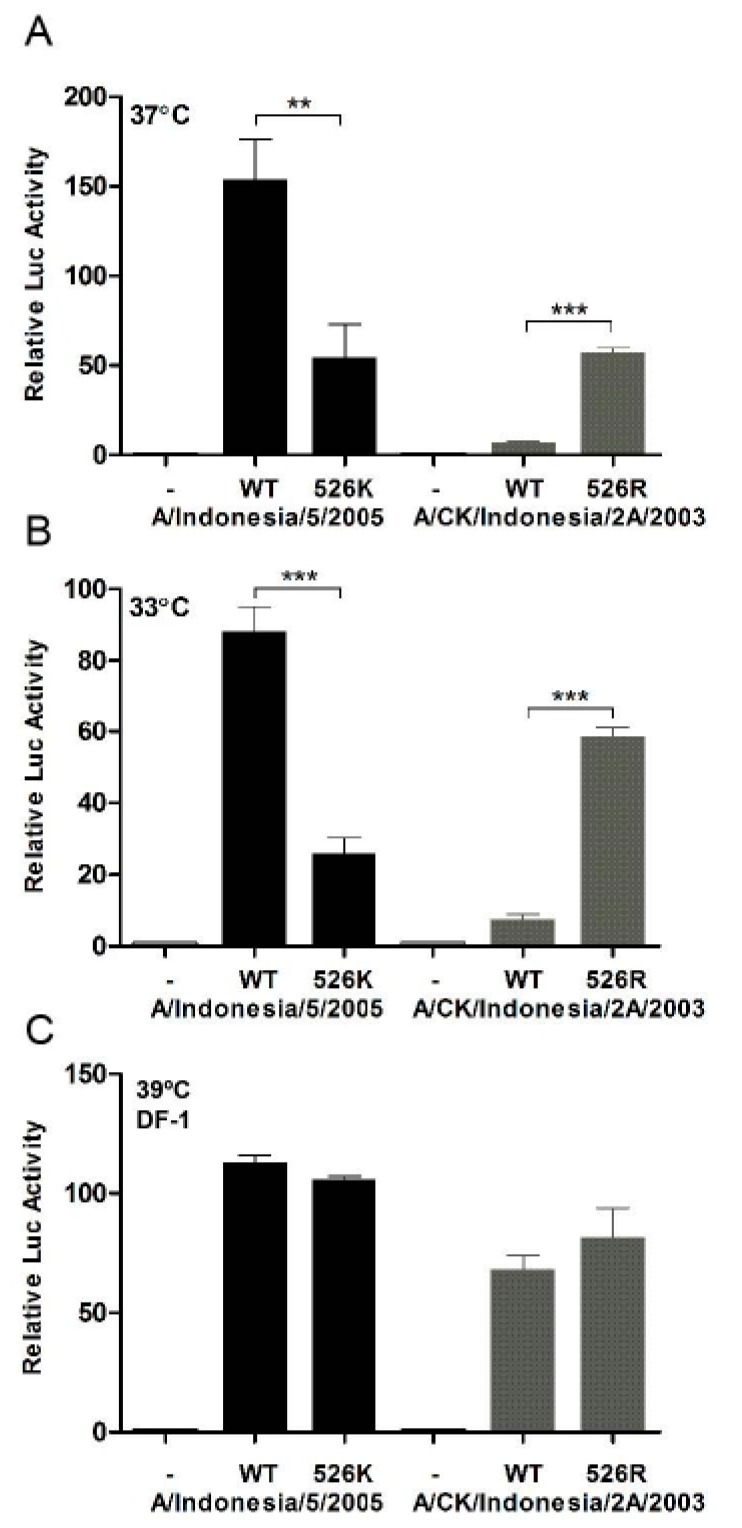
RNP polymerase activity in HEK293T and DF-1 cells. RNP expression plasmids containing NP, PB1, PA and wild type PB2, or the corresponding PB2 526 mutant derived from CK2A or IND5, along with a firefly luciferase RNP reporter plasmid and the Renilla luciferase expressing plasmid, pRL-TK (as an internal control), were transfected into HEK293T cells. The transfected cells were incubated for 24 h at: 37 °C (**A**); or 33 °C (**B**). DF-1 cells were transfected with the same sets of RNP complex plasmids but with an avian specific RNP firefly reporter and Renilla luciferase control (**C**). DF-1 cells were incubated at 39 °C. Luciferase activities were measured at 24 h post transfection using a Dual-Luciferase Reporter Assay System (Promega). Firefly RNP activity was normalized against Renilla activity. RNP without PB2 was used as a negative control. Data represent mean normalized luciferase activity from three independent experiments. Error bars represent standard deviation calculated from three separate experiments. Statistical significance was analyzed by Student’s *t*-test. *** *p* < 0.001, ** *p* < 0.01.

**Figure 3 viruses-11-00292-f003:**
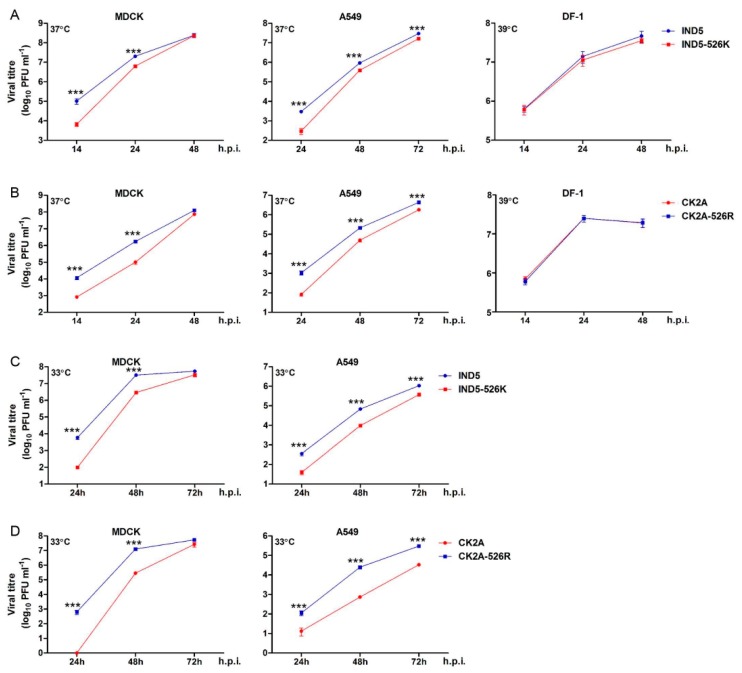
Growth kinetics of H5N1 viruses in mammalian and avian cells. Reverse genetic (RG) versions of the human H5N1 isolate, A/Indonesia/5/05 (IND5), and a mutant virus containing 526K-PB2 (IND5-526K) were used to infect MDCK and DF-1 cells at an MOI of 0.001, or A549 cells at an MOI of 0.01 (**A**). Similarly, RG versions of the avian H5N1 isolate, A/Chicken/Indonesia/2A/03 (CK2A), and a mutant virus containing 526R-PB2 (CK2A-526R) were used to infect MDCK and DF-1 cells at an MOI of 0.001, or A549 cells at an MOI of 0.01 (**B**). Growth of IND5 (**C**) and CK2A (**D**) viruses at 33 °C was estimated in MDCK and A549 cells. After adsorption for 1 h, the cells were washed and overlaid with infection medium containing 1 µg/mL of TPCK-trypsin. At the indicated time points, culture supernatants were collected for virus titration by plaque assay in MDCK cells. Data represent mean viral titers from three independent experiments. Error bars represent standard deviation calculated from three separate experiments. Statistical significance was analyzed by one-way ANOVA, corrected by the Bonferroni post test: *** *p* < 0.001., h.p.i., h post infection.

**Figure 4 viruses-11-00292-f004:**
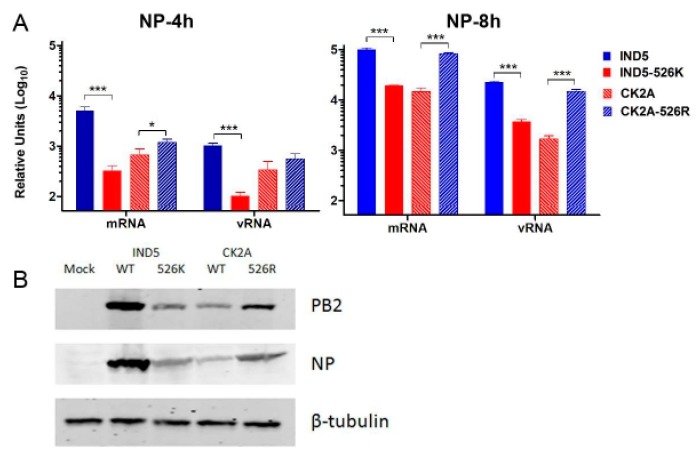
Effect of PB2 526R on H5N1 viral RNA and protein synthesis. (**A**) A549 cells were infected with reverse genetic wild type IND5 or CK2A H5N1 viruses or mutant versions of these viruses containing either 526K or 526R PB2, respectively, at an MOI of 1. Total RNA was extracted at 4 or 8 h post infection and reverse transcription (RT) performed using uni-12 or oligo dT primers, to detect vRNA or mRNA, respectively. Expression of these two RNA species of NP was quantified by relative quantitative real time RT-PCR and normalized against the β-actin gene. Data represent mean relative NP expression levels from three independent experiments. Statistical significance was analyzed by one-way ANOVA, corrected by the Bonferroni post test: *** *p* < 0.001and * *p* < 0.05. (**B**) A549 cells were infected as above. Cells were lysed at 8 h post infection and expression levels of NP and PB2 estimated by Western blot using anti-NP (1:5000) and anti-PB2 (1:1000) antibodies. Expression levels of β-tubulin were detected using anti-β-tubulin antibody, and served as a loading control.

**Figure 5 viruses-11-00292-f005:**
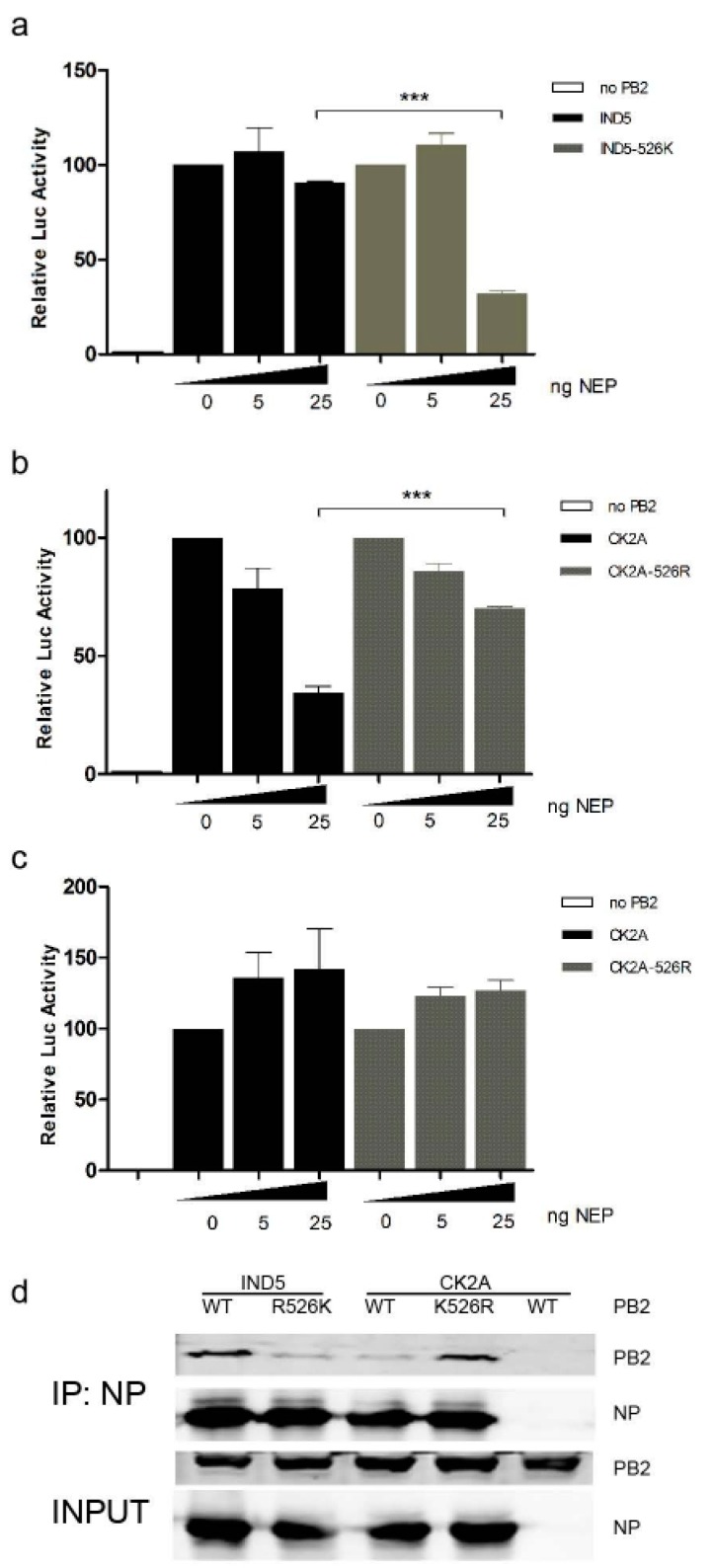
Effect of 526R PB2 on RNP activity in the presence of NEP and NP-PB2 interaction. Increasing amounts of NEP expression vector, or empty vector, were co-transfected with expression plasmids for the RNP complexes of IND5 and IND5-526K PB2 (**A**), or CK2A and CK2A-526R PB2 (**B**), together with RNP luciferase reporter and pRL-TK control plasmids, into HEK293T cells, followed by culture at 37 °C. Similarly, RNP polymerase activity with the RNP complexes of CK2A and CK2A-526R and increasing amounts of NEP expression was examined in DF-1 cells, cultured at 39 °C (**C**). Luciferase activities were measured at 24 h post transfection. Firefly RNP polymerase activity was normalized against Renilla activity, and RNP without PB2 was used as a negative control. Data represent mean normalized luciferase activity from three independent experiments. Error bars represent standard deviation from three separate experiments. Statistical significance was analyzed by Student’s *t*-test. *** *p* < 0.001. (**D**) Interaction between NP and PB2 in RNP complexes. Different sets of plasmids expressing PB1, PA, NP and Flag-tagged PB2 from IND5-526K or CK2A-526R, were transfected into HEK293T cells. At 48 h post transfection, cell lysates were prepared and rabbit polyclonal NP antibody used to co-precipitate PB2. Proteins from co-precipitated complexes were resolved using SDS-PAGE and detected by Western blot using anti-NP (1:5000) and anti-Flag (Sigma) (1:5000) antibodies.

**Figure 6 viruses-11-00292-f006:**
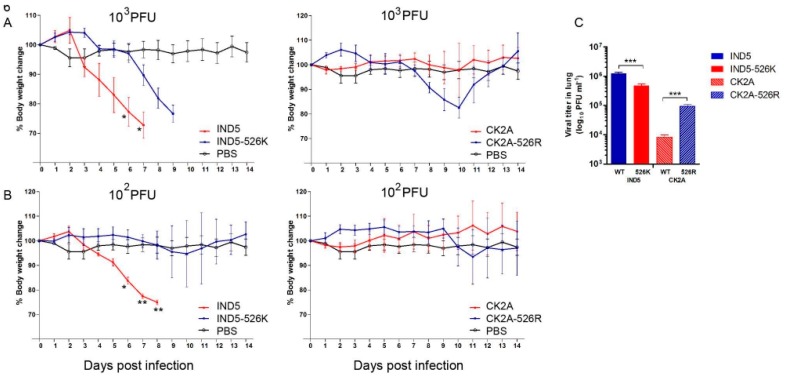
Evaluation of H5N1 virus infection and replication in mice. Groups of four or six BALB/c mice, aged 4–6 weeks, were intranasally inoculated with 10^3^ (**A**) or 10^2^ (**B**) PFU of virus containing wild type IND5 or CK2A, or the 526K (IND5-526K) or 526R (CK2A-526R) mutant viruses, in 25ul PBS. Body weight and survival were monitored daily for 14 days post infection. Results of infection with high doses of virus (10^4^, 10^5^ and 10^6^ PFU) are shown in the Appendix A. The MLD_50_ was calculated by the method of Reed and Muench [30]. (**C**) Replication efficiency of viruses in lung tissues of infected mice. To determine virus titers in mouse lung tissues, groups of three mice were infected with 10^3^ PFU of the respective viruses described above and then euthanized at 72 h post infection, with lung tissues from each mouse being collected and homogenized for virus titration by plaque assay using MDCK cells. Error bars represent standard deviation from virus-infected mice mouse in the group. Statistical significance was analyzed by one-way ANOVA or Student’s *t*-test *** *p* < 0.001, ** *p* < 0.01 and * *p* < 0.05.

**Figure 7 viruses-11-00292-f007:**
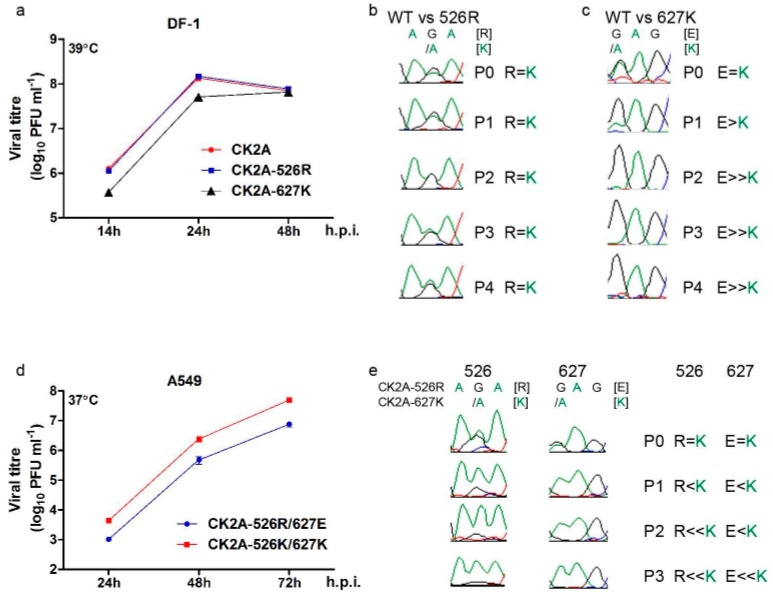
Comparison of effects of PB2 526R and 627K on Indonesian H5N1 virus replication. (**A**) Growth kinetics of reverse genetic versions of CK2A virus containing PB2 526R or 627K in avian cells. DF-1 cells were infected separately with CK2A, CK2A-526R or CK2A-627K RG virus and cultured at 39 °C. Culture supernatant was collected at the indicated time points and virus titrated by plaque assay in MDCK cells. (**B**) Growth competition assay: CK2A wild type versus CK2A-526R virus in avian cells. DF-1 cells were infected at an MOI of 0.001 with CK2A and CK2A-526R viruses, mixed at a ratio of 1:1. Continuous cultures were performed for four passages. Viral RNA was isolated from culture supernatants at 48 h post infection, and the PB2 gene amplified by RT-PCR and sequenced. Representative sequencing chromatograms from each passage are displayed; the genetic code for PB2 526R is AGA, whereas 526K is AAA. (**C**) Growth competition assay: CK2A wild type versus CK2A-627K virus in avian cells. The assay was conducted similarly to (**B**). Representative sequencing chromatograms are shown: the genetic code for PB2 627E is GAG, whereas 627K is AAG. (**D**) Growth kinetics of reverse genetic versions of CK2A virus containing 526R or 627K. A549 cells were infected separately with RG CK2A-526R or CK2A-627K viruses and cultured at 37 °C. Culture supernatant was collected at the indicated time points and virus titrated by plaque assay in MDCK cells. (**E**) Growth competition assay: CK2A-526R versus CK2A-627K virus in A549 cells. The procedure was performed as in (**B**). Sequencing chromatograms representing mixed populations of 526R/K and 627E/K are shown. h.p.i., hours post infection. Error bars represent standard deviation from three separate experiments. Statistical significance was analyzed by one-way ANOVA or Student’s *t*-test *** *p* < 0.001.

**Figure 8 viruses-11-00292-f008:**
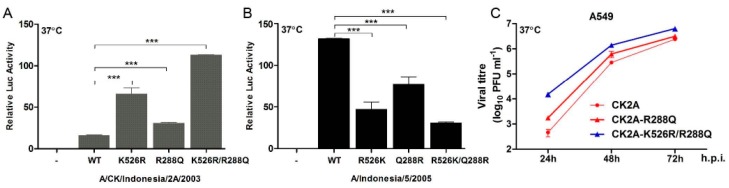
Effect of PB2 R288Q on viral replication and RNP activity in Indonesian H5N1 viruses. R288Q, alone and in combination with K526R, was introduced into the PB2 segment of the H5N1 avian isolate, A/Chicken/Indonesia/2A/2003, and alternatively, the back mutations Q288R and R526K, separately and in combination, were introduced into the PB2 of the H5N1 human isolate, A/Indonesia/5/2005. Plasmids of the minigenome system, consisting of NP, PB1, PA and wild type PB2, or the corresponding PB2 mutants derived from CK2A or IND5, together with a firefly RNP luciferase reporter plasmid and the Renilla luciferase expressing plasmid pRL-TK (as an internal control) were transfected into HEK293T cells. (**A**) Effect of R288Q, alone or in conjunction with K526R, on RNP polymerase activity in the background of other RNP proteins from A/Chicken/Indonesia/2A/2003. (**B**) Effect of Q288R, alone or in conjunction with R526K, on RNP polymerase activity, with other RNP proteins derived from A/Indonesia/5/2005. Luciferase activities were measured at 24 h post transfection using a Dual Luciferase Reporter Assay System (Promega). Firefly RNP activity was normalized against Renilla activity. RNP without PB2 was used as a negative control. Data represent mean normalized luciferase activity from three independent experiments. Statistical significance was analyzed by Student’s *t*-test. *** *p* < 0.001. (**C**) Growth kinetics of wild type CK2A virus and mutant viruses containing R288Q PB2 or R288Q/K526R PB2. A549 cells were infected with CK2A, CK2A-R288Q or CK2A-K526R/R288Q viruses at an MOI of 0.01 and cultured at 37 °C. Culture supernatants were collected at the indicated time points and virus titrated by plaque assay in MDCK cells. Error bars represent standard deviation from three separate experiments.

**Table 1 viruses-11-00292-t001:** Determination of the MLD_50_ of RG viruses*.

Viral Strain	MLD_50_ (PFU)
IND5 WT	1.5
IND5-526K	3.2 × 10^2^
CK2A	2.2 × 10^3^
CK2A-526R	2.2 × 10^3^

* Groups of four mice were inoculated with different doses of virus, ranging from 1 to 10^6^ PFU, and mouse survival monitored for 14 days post infection. The MLD_50_ was calculated by the method of Reed and Muench [30].

**Table 2 viruses-11-00292-t002:** E627K adaptive mutations in mice fatally infected with H5N1 virus *.

Viral Strain	Number of Mice	Mouse Group (Infective Viral Dose, PFU) in Which E627K Mutation Emerged
IND5 WT	0/7	None detected
IND5-526K	2/4	10^3^
CK2A WT	3/4	One from 10^2^
		Two from 10^3^
CK2A-526R	1/5 ^&^	10^4^

***** Virus isolated from fatally infected mice died or euthanized from Day 4 post infection was characterized for adaptation markers in the PB2 gene by sequencing analysis. ^&^ Among the five fatally infected mice in this group, only one mouse was found to contain E627K PB2 and K526R remained unchanged in this mouse.

**Table 3 viruses-11-00292-t003:** Comparison of amino acid variations in the PB2 segments of the IND5 and CK2A strains.

Viral Strain	Position in PB2 Gene							
	288	323	368	389	524	526	658	756
IND5	**Q**	L	R	R	I	R	Y	M
CK2A	**R**	F	Q	K	T	K	H	T

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
