# Peer review of "The PB2 Polymerase Host Adaptation Substitutions Prime Avian Indonesia Sub Clade 2.1 H5N1 Viruses for Infecting Humans"

_viruses, 2019, doi:10.3390/v11030292_

Reviewer 1 Report

In this manuscript Wang and colleagues test the hypothesis that PB2 K526R  primes Indonesian avian H5N1 viruses (clade 2.1) for human infection. In a previous study the authors provide data that “support the notion that K526R is a new adaptation marker for H7N9 and H5N1-(Indonesia) avian influenza A virus in humans”. This work is a more elaborate evaluation of the effects of PB2 K526R in Indonesian H5N1 clade 2.1 viruses. The results from the current study support the claims made by the authors, although the conclusions from the abstract and main text should be phrased more carefully in my opinion to prevent overstating the outcomes of this work. I have a few suggestions that I think could improve the manuscript.

The title of the manuscript is too general and should mention PB2 position 526 and also that this work was done exclusively using Indonesian H5N1 clade 2.1 viruses.

 The data included in Fig. 1 seem much more limited than what is currently available from sequence databases and include very few strains from 2007 and later. What is the amino acid at PB2 position 526 for the 107 human isolates and 101 avian isolates currently available from GISAID? The lack of information about PB2 526 in recent viruses makes it harder to translate the results from this study into actions. Additional phylogenetic analyses on PB2 526R/K in recent strains is needed to make this work relevant to the suggestion of the authors that PB2 position 526 should be screened to identify viruses with zoonotic potential.

 The human isolates with PB2 526R are in a different clade than most of the avian viruses with PB2 K526R. What are the genetic changes responsible for this difference. Also, the authors should exclude that the important phylogenetic differences (526R vs 526K, and human 526R vs avian 526R) are due to differences in the way that these viruses were propagated (eggs, MDCKs, MDCK-SIAT1, etc)?

The authors associate the higher number of human infections in Indonesia with PB2 526R. This association requires an analysis of PB2 526R in other H5N1 clades than 2.1 to show that 526R is indeed absent/rare in regions where the number of human cases is lower than in Indonesia.

 Line 313: “[…], these results clearly show that K526R substitution in PB2 significantly enhances virus replication in mammalian cells, […]”. This line should state that the K526R substitution enhances replication in the cells used in the experiments described in this manuscript.

 Line 332: “[…], supporting the idea that the K526R substitution is critical for infection of mice with Indonesian H5N1 viruses”. This statement is not correct. As shown in Fig. 6C 526K viruses do replicate in mice. Please adjust to state that 526R enhances replication.

 Line 379: “[…], which suggests that 526R PB2 has no fitness cost […]. Fitness costs may be very subtle and hard to measure. Therefore, the results of the analysis done for this manuscript do not show that 526R PB2 has no fitness cost, but merely that a fitness cost was not detected with the method used here. Please adjust the text accordingly.

 Line 384: I think the conclusion from line 384 is an overinterpretation of the results from this work and should be rephrased. Suggest to change to: Taken together, these results indicate that host adaptive mutations in the PB2 gene enhance H5N1 virus replication in mice and that K526R PB2 is a pre-existing adaptive marker present in Indonesian H5N1 viruses currently circulating in avian species which supports increased virus replication in mammalian cells.

Line 426: The authors state that “[…], the K526R PB2 substitution in Indonesian H5N1 viruses is rarely associated with either E627K or D701N”, which is not supported by data provided in the manuscript. A more extensive and updated phylogenetic analysis like suggested above could support this statement.

 It is surprising that the analysis from Fig. 7 was performed only in A549 as representative for mammalian cells, and not in MDCKs like was done for the analysis in from Fig. 3. Please clarify this choice.

 Line 441: replace indicate by suggest.

 Line 81: “[…], it seems possible that for the genetic constellation of Indonesian H5N1 viruses optimal function of the polymerase complex for virus replication is independent of E627K, and instead requires K526R adaptive substitution”. There is no indication that K526R is required as an adaptive solution, there may be other adaptive markers that happened to not get selected during evolution. Change requires to utilizes.

 Line 44: There is a space in the URL

 Line 50: There is an extra space between emergence and of

Author Response

Response to reviewers

Reviewer 1

Comments and Suggestions for Authors

In this manuscript Wang and colleagues test the hypothesis that PB2 K526R  primes Indonesian avian H5N1 viruses (clade 2.1) for human infection. In a previous study the authors provide data that “support the notion that K526R is a new adaptation marker for H7N9 and H5N1-(Indonesia) avian influenza A virus in humans”. This work is a more elaborate evaluation of the effects of PB2 K526R in Indonesian H5N1 clade 2.1 viruses. The results from the current study support the claims made by the authors, although the conclusions from the abstract and main text should be phrased more carefully in my opinion to prevent overstating the outcomes of this work. I have a few suggestions that I think could improve the manuscript.

The title of the manuscript is too general and should mention PB2 position 526 and also that this work was done exclusively using Indonesian H5N1 clade 2.1 viruses.

 Response:  Title has been modified in revised version

The data included in Fig. 1 seem much more limited than what is currently available from sequence databases and include very few strains from 2007 and later. What is the amino acid at PB2 position 526 for the 107 human isolates and 101 avian isolates currently available from GISAID? The lack of information about PB2 526 in recent viruses makes it harder to translate the results from this study into actions. Additional phylogenetic analyses on PB2 526R/K in recent strains is needed to make this work relevant to the suggestion of the authors that PB2 position 526 should be screened to identify viruses with zoonotic potential.

The human isolates with PB2 526R are in a different clade than most of the avian viruses with PB2 K526R. What are the genetic changes responsible for this difference. Also, the authors should exclude that the important phylogenetic differences (526R vs 526K, and human 526R vs avian 526R) are due to differences in the way that these viruses were propagated (eggs, MDCKs, MDCK-SIAT1, etc)?

Response: Fig.1 has been modified to include update H5N1 virus sequences from Indonesia

The authors associate the higher number of human infections in Indonesia with PB2 526R. This association requires an analysis of PB2 526R in other H5N1 clades than 2.1 to show that 526R is indeed absent/rare in regions where the number of human cases is lower than in Indonesia.

Response: Our previous publication (Song W et al., Nat Commun 2014)  has shown that K526R is prevalent in Indonesia but not in other subclade of H5N1 virus.

Line 313: “[…], these results clearly show that K526R substitution in PB2 significantly enhances virus replication in mammalian cells, […]”. This line should state that the K526R substitution enhances replication in the cells used in the experiments described in this manuscript.

 Response: Modification has been made in the revised version.

Line 332: “[…], supporting the idea that the K526R substitution is critical for infection of mice with Indonesian H5N1 viruses”. This statement is not correct. As shown in Fig. 6C 526K viruses do replicate in mice. Please adjust to state that 526R enhances replication.

 Response: Modification has been made.

Line 379: “[…], which suggests that 526R PB2 has no fitness cost […]. Fitness costs may be very subtle and hard to measure. Therefore, the results of the analysis done for this manuscript do not show that 526R PB2 has no fitness cost, but merely that a fitness cost was not detected with the method used here. Please adjust the text accordingly.

 Response: We agree with reviewer comment and have adjust the text in the revised version.

Line 384: I think the conclusion from line 384 is an overinterpretation of the results from this work and should be rephrased. Suggest to change to: Taken together, these results indicate that host adaptive mutations in the PB2 gene enhance H5N1 virus replication in mice and that K526R PB2 is a pre-existing adaptive marker present in Indonesian H5N1 viruses currently circulating in avian species which supports increased virus replication in mammalian cells.

 Response: Modification has been made to address the concern.

Line 426: The authors state that “[…], the K526R PB2 substitution in Indonesian H5N1 viruses is rarely associated with either E627K or D701N”, which is not supported by data provided in the manuscript. A more extensive and updated phylogenetic analysis like suggested above could support this statement.

 Response: Additional reference has been added in the revised version to address this question.

It is surprising that the analysis from Fig. 7 was performed only in A549 as representative for mammalian cells, and not in MDCKs like was done for the analysis in from Fig. 3. Please clarify this choice.

Response: MDCK cells are permissive to infection of most of avian influenza viruses while A549 cells show restriction to some avian viruses. We considered that A549 cells is more suitable for experiment to test replication efficiency of PB2 adaptive mutations of H5N1 virus. 

Line 441: replace indicate by suggest.

Response: Modification has been made.

Line 81: “[…], it seems possible that for the genetic constellation of Indonesian H5N1 viruses optimal function of the polymerase complex for virus replication is independent of E627K, and instead requires K526R adaptive substitution”. There is no indication that K526R is required as an adaptive solution, there may be other adaptive markers that happened to not get selected during evolution. Change requires to utilizes.

Response: We have modified the text in the revised version.

Line 44: There is a space in the URL

 Response: Correction has been made.

Line 50: There is an extra space between emergence and of

Response: Correction has been made.

Reviewer 2 Report

In this manuscript by Pui Wang et al., the authors focused on a substitution  (K526R) in the influenza virus polymerase subunit PB2, a substitution that  was identified in Indonesian isolates shortly after the emergence of H5N1  viruses in Indonesia, and that was suggested to be associated with  replication potential in mammalian cells.The goal of this study was therefore to examine whether the K526R  substitution in PB2 predisposes the virus for human infection.

The main findings are:-          526R-PB2 became predominant in Indonesian H5N1 isolates (avian and human) from ~2005-          Two indonesian H5N1 viruses were rescued by reverse-genetics: avian virus CK2A (with PB2-526K), and human virus IND5 (with PB2-526R).-          A minireplicon assay in HEK293T cells showed that the IND5-derived viral polymerase [526R] was much more active than the CK2A-derived polymerase [526K]. In addition, substitution PB2-R526K in IND5-derived polymerase considerably decreased its activity, while reciprocally that of the CK2A-polymerase was increased by the PB2-K526R substitution. This demonstrated that in this context, R526 increased the activity of the viral polymerase. The activity ratios 526R/526K or 526K/526R were even more pronounced when assayed at 33°C.-          By contrast the substitutions K526R (in CK2A) and R526K (in IND5) had a much milder impact on the viral polymerase activity when this was assayed in avian DF-1 cells.-          Similar results were found when assaying viral replication in mammalian (MDCK, A549) or avian (DF-1) cells: R526 was associated with a higher replication potential in mammalian cells.-          Viruses harbouring PB2-R526 (i.e. IND5 and CK2A-K526R) also showed increased pathogenicity in mice-          Although the K526R substitution in PB2 is a potent determinant of mammalian adaptation, other substitutions, notably R288Q in PB2, may also be involved. 
The manuscript is interesting and deals with an important issue regarding adaptation to mammals of avian influenza viruses. However, there are a number of points that need to be addressed by the authors, including some inconsistencies in the lethality for mice. 
Major remarksFigure 2. What exactly is the origin of the PB1, PA and NP subunits of the polymerase, IND5 or CK2A? (see also remark for lines 427-443 and to Fig 8). 
In Figure 4A, there is some ambiguity about the values and the y-scale. To remove that ambiguity, and for the sake of readability, values (not their log10) should be reported, and the log10 y-scale should be emphasized by its subdivisions [2-9] between powers of 10. Same remark for Fig 6C. 
Lines 286-87. Between “We tested…” and “Our results show…”, a sentence should explain the rationale of the test being used (what is the rationale of adding increasing amounts of NEP?). This would also help in understanding the data (lines 288-92). Same remark for the PB2-NP interaction (lines 292-95). In brief, how NEP and NP could be expected to modulate the activity of the viral polymerase? Further, these possible interactions with NEP and NP are not discussed at all (neither here nor in the discussion section). 
Line 327 and Table 1. Contradictory data. “mice infected with 10^3 PFU of the avian isolate CK2A showed no disease symptoms”, while according to Table 1, the MLD50 of CK2A is 2.2 x 10^3 PFU. Same inconsistency for CK2A-526R (all mice survived with an inoculum of 10^3, but MLD50 is 2.2x 10^3). Moreover, according to the legend to Table 1, MLD50 were calculated after inoculating mice with 10^2 to 10^6 PFU. With such a protocol, it is impossible to reliably calculate an MLD50 below 10^3. 
Fig 7E is somewhat misleading, in that one may understand that there is a competition between four viruses (526 R or K and 627 E or K). It would be better to put the name of the two viruses in front of the nucleotide sequence (CK2A-526R in front of AGA [R] ..GAG [E], and CK2A-627K in front of the line below. The data show that the latter “wins” the competition. The legend also is misleading. Line 417-18 should read “..mixed populations of [526R-627E] and [526K-627K]”.Line 383… rather “Comparisons of the two CK2A-derived viruses (i.e. [526R-627E] and [526K-627K]) through either virus growth kinetics or competition assays suggested that the E627K substitution had a greater positive effect on the replication potential in A549 cells than the K526R substitution”. 
Lines 427-443, 117-119 and 109-110. The available nucleotide sequences of the CK2A virus are only partial, precluding full-length alignments. Nevertheless, comparison of the available PB2 sequences shows another substitution (G669V) between IND5 and CK2A. Furthermore, comparison of the other polymerase subunit sequences reveals at least 2 substitutions between the PB1s and 4 substitutions between the PAs. But in fact sometimes it is difficult to find the precise information  about the composition of the polymerase complex in the minireplicon assays (what is the origin of the PB1, PA and NP subunits, IND5 or CK2A?). 
More specifically for lines 427-443 and Fig. 8: since there are other substitutions (not only in PB2, but also in PB1 and PA) between IND5 and CK2A that could subtly alter the activity of the polymerase, it is necessary to know the relative efficiency of the two polymerase complexes (IND5 and CK2A). To this end, Graphs A and B in Fig. 8 should be combined in a unique graph, with the 100% value assigned to IND5-wt (otherwise, the activity of the wt-CK2A RNP relative to that of IND5 could be added in Fig 8A). Maybe a logarithmic y-scale (log2 or log10) could help to clearly see the differences between the conditions. 
Lines 441-43. It is somewhat excessive to assign the difference of polymerase activity solely to the two PB2 substitutions R288Q and K526R, since (i) all the substitutions were not assayed and (ii) other substitutions in PB1 and PA may also play a role. 
Lines 535-39. Perhaps a sentence could be added with the meaning that there is more than one unique pathway of adaptation to mammals for the viral polymerase, since at least two substitutions [PB2-K526R and PB2-E627K] have now been identified that prime these viruses for infecting human; with the added information that K526R does not negatively impact the viral replication in avian cells. 
Perhaps the authors should try to map the substitutions in the structure of the viral polymerase (cf Pflug 2014, Pflug 2018), at least to the main structural domains of PB2. 

Minor remarksLine 21. ..human infection, we showed that: (1)…Lines 43-44. Please update the figures (as to September 2017, there were 860 lab-confirmed infections including 454 deaths).Line 208. Somewhat ambiguous. Maybe rather “…the later-emerging avian H5N1 viruses which carry 526R PB2 (in blue)….Line 285….that 627K and 526R substitutions in PB2Line 366… E627K substitution was [] found in one of five mice…(the meaning of “only” is ambiguous here).Line 482. Initiated by the [] emergence…(sudden may be excessive: cf ref 44, which should be added here with 42-43).Line 533. …primarily..

Author Response

Reviewer 2

Comments and Suggestions for Authors

Major remarks

Figure 2. What exactly is the origin of the PB1, PA and NP subunits of the polymerase, IND5 or CK2A? (see also remark for lines 427-443 and to Fig 8). 

Response:

 In Figure 4A, there is some ambiguity about the values and the yscale. To remove that ambiguity, and for the sake of readability, values (not their log10) should be reported, and the log10y scale should be emphasized by its subdivisions [29] between powers of 10. Same remark for Fig 6C. 

 Response: Fig. 4A and Fig. 6C have been modified in the revised version.

Line28687. Between “We tested…” and “Our results show…”, a sentence should explain the rationale of the test being used (what is the rationale of adding increasing amounts of NEP?). This would also help in understanding the data (lines 288-92). Same remark for the PB2-NP interaction (lines 292 95). In brief, how NEP and NP could be expected to modulate the activity of the viral polymerase? Further, these possible interactions with NEP and NP are not discussed at all (neither here nor in the discussion section). 28687                                                                                                                        

Response: We have modified text and included additional reference to state the rationale of this experiment.

 Line 327 and Table 1. Contradictory data. “mice infected with 10^3 PFU of the avian isolate CK2A showed no disease symptoms”, while according to Table 1, the MLD50 of CK2A is 2.2 x 10^3 PFU. Same inconsistency for CK2A-526R (all mice survived with an inoculum of 10^3, but MLD50 is 2.2x 10^3). Moreover, according to the legend to Table 1, MLD50 were calculated after inoculating mice with 10^2 to 10^6 PFU. With such a protocol, it is impossible to reliably calculate an MLD50 below 10^3. 

 Response The challenge dose for different groups of mice is from 1-106 pfu per mice (as shown in methods).  For CK2A and CK2A-K526R groups, no mice died when the dose is equal or lower than 10^3 pfu. Therefore, the MLD50 of some groups is lower. Table 1 legend has a typo, challenge dose should to 1-106 pfu. We have made correction in the revised version.

Fig 7E is somewhat misleading, in that one may understand that there is a competition between four viruses (526 R or K and 627 E or K). It would be better to put the name of the two viruses in front of the nucleotide sequence (CK2A-526R in front of AGA [R] ..GAG [E], and CK2A-627K in front of the line below. The data show that the latter “wins” the competition. The legend also is misleading. Line 417-18 should read “..mixed populations of [526R-627E] and [526K-627K]”.Line 383… rather “Comparisons of the two CK2A-derived viruses (i.e. [526R-627E] and [526K-627K]) through either virus growth kinetics or competition assays suggested that the E627K substitution had a greater positive effect on the replication potential in A549 cells than the K526R substitution”. 

 Response: We have modified Fig. 7E and revised text to address these concerns

 Lines 427-443, 117-119 and 109-110. The available nucleotide sequences of the CK2A virus are only partial, precluding full-length alignments. Nevertheless, comparison of the available PB2 sequences shows another substitution (G669V) between IND5 and CK2A. Furthermore, comparison of the other polymerase subunit sequences reveals at least 2 substitutions between the PB1s and 4 substitutions between the PAs. But in fact sometimes it is difficult to find the precise information  about the composition of the polymerase complex in the minireplicon assays (what is the origin of the PB1, PA and NP subunits, IND5 or CK2A?). 

Response: CK2A was used because it was isolated during 2003 before H5N1 human cases emerged in Indonesia. Residues selected for testing in this study on available sequences. We understand these residues tested can not fully represent all the possibilities. Further study based on analyses of  more sequences of all PB2 with K526R substitution will provide more information for understand role of other adaptive mutation associated with K526R \.

 More specifically for lines 427-443 and Fig. 8: since there are other substitutions (not only in PB2, but also in PB1 and PA) between IND5 and CK2A that could subtly alter the activity of the polymerase, it is necessary to know the relative efficiency of the two polymerase complexes (IND5 and CK2A). To this end, Graphs A and B in Fig. 8 should be combined in a unique graph, with the 100% value assigned to IND5-wt (otherwise, the activity of the wt-CK2A RNP relative to that of IND5 could be added in Fig 8A). Maybe a logarithmic y-scale (log2 or log10) could help to clearly see the differences between the conditions. 

 Lines 441-43. It is somewhat excessive to assign the difference of polymerase activity solely to the two PB2 substitutions R288Q and K526R, since (i) all the substitutions were not assayed and (ii) other substitutions in PB1 and PA may also play a role. 

Response: We agree with this reviewer that there are other possibilities in PB1 and PA of both strains. Fig. 8 tests if R288Q is associated with K526R for enhanced polymerase activity in Indonesia H5N1 virus. We consider it is easier for reader to understand acquired mutation,R288Q, would  enhance PB2 K526R RNP polymerase activity in the A/CK/Indonesia/2A/03 background (PB2-526K) while reversed mutation, Q288R, would negatively affect RNP polymerase activity in the A/Indonesia/05/03 background (PB2-526R).

Lines 535-39. Perhaps a sentence could be added with the meaning that there is more than one unique pathway of adaptation to mammals for the viral polymerase, since at least two substitutions [PB2-K526R and PB2-E627K] have now been identified that prime these viruses for infecting human; with the added information that K526R does not negatively impact the viral replication in avian cells. 
Perhaps the authors should try to map the substitutions in the structure of the viral polymerase (cf Pflug 2014, Pflug 2018), at least to the main structural domains of PB2. 

 Response: We have modified text in the revised version to state multiple adaptive strategies are utilized by different avian influenza virus to gain cross species replication.

Minor remarksLine 21. ..human infection, we showed that: (1)…Lines 43-44. Please update the figures (as to September 2017, there were 860 lab-confirmed infections including 454 deaths).Line 208. Somewhat ambiguous. Maybe rather “…the later-emerging avian H5N1 viruses which carry 526R PB2 (in blue)….Line 285….that 627K and 526R substitutions in PB2Line 366… E627K substitution was [] found in one of five mice…(the meaning of “only” is ambiguous here).Line 482. Initiated by the [] emergence…(sudden may be excessive: cf ref 44, which should be added here with 42-43).Line 533. …primarily..

 Response: Corrections have been made in the revised version 

Round  2

Reviewer 2 Report

Comments on the revised manuscript :

As for Figure 4A and 6C, the authors probably misunderstood my remark.  In fact the first version of these figures was OK with the y-scale  representing powers of ten. I simply suggested to add minor ticks  representing the subdivisions 2 to 9 between powers of ten, so as to  emphasize the logarithmic scale (in GraphPad, just click on  menu “format axes” and chose “minor ticks”).

Lines 293-307 and discussion. The authors could devote at least a  line or a sentence of the discussion section to the interactions of PB2  with NEP and NP.

Minor remarksLine 22. We previously showed  substitution PB2-K526R is present in 80%... 
Line 295. ..functions of …

Author Response

As for Figure 4A and 6C, the authors probably misunderstood my remark.  In fact the first version of these figures was OK with the y-scale  representing powers of ten. I simply suggested to add minor ticks  representing the subdivisions 2 to 9 between powers of ten, so as to  emphasize the logarithmic scale (in GraphPad, just click on  menu “format axes” and chose “minor ticks”).

Response: We apologized for misunderstood this reviewer's comment in previous response. Revision has been made in current revised version.

Lines 293-307 and discussion. The authors could devote at least a  line or a sentence of the discussion section to the interactions of PB2  with NEP and NP.

Response: We agree with this reviewer's comment and have included a discussion to integrate PB2 with NEP and NP and added additional references in the revised version. 

Minor remarksLine 22. We previously showed  substitution PB2-K526R is present in 80%... 
Line 295. ..functions of …

Response: Corrections have been made.